# MCTransformer: Combining Transformers and Monte-Carlo Tree Search for Offline Reinforcement Learning

## Abstract

Recent studies explored the framing of reinforcement learning as a sequence modeling problem, and then using Transformers to generate effective solutions. In this study, we introduce *MCTrasnformer*, a framework that combines Monte-Carlo Tree Search (MCTS) with Transformers. Our approach uses an actor-critic setup, where the MCTS component is responsible for navigating previously-explored states, aided by input from the Transformer. The Transformer controls the exploration and evaluation of new states, enabling an effective and efficient evaluation of various strategies. In addition to the development of highly effective strategies, our setup enables the use of more efficient sampling compared to existing MCTS-based solutions. *MCTrasnformer*, therefore, is able to effectively learn from a small number of simulations for each newly-explored node, without degrading performance. Evaluation conducted on the challenging and well-known problem of SameGame, shows that *MCTrasnformer* outscores Transformer-only and MCTS-only solutions by a factor of three and more.

## 1 Introduction

Transformers have recently been shown to be very effective in the field of reinforcement learning (RL) Chen et al. (2021); Janner et al. (2021). This was achieved by converting offline RL to a classification problem, which facilitates advanced sequence modeling abilities using Transformers. At evaluation time, the Transformer functions as an autoregressive model, generating sequences of future actions.

The main shortcoming of the aforementioned approach is the absence of exploration during online evaluation. The Transformer model is, therefore, limited in its ability to adjust to novel circumstances. While online fine-tuning of the model was recently proposed Zheng et al. (2022a), it requires a relatively large number of training samples. Moreover, a one-time fine-tuning approach is not suitable for problems with high degrees of volatility, that require continuous exploration.

We introduce *MCTrasnformer*, a RL framework that enables cost-effective exploration of planning problems. Our approach combines Monte-Carlo Tree Search (MCTS) with the Transformer architecture in an actor-critic setup. The MCTS component of *MCTrasnformer* is tasked with balancing the exploration/exploitation trade-off needed in most RL tasks, while the Transformer component is tasked with predicting the utility of previously-unexplored nodes. Additionally, we use the Transformer to carry out the simulation phase of the MCTS (i.e., the rollout policy), where the former's advanced and effective modeling allows us to use a vary small number of simulations, thus keeping our approach efficient.

We evaluate the proposed approach on SameGame, a challenging and well-known problem. This game is considered challenging due to the high variance of its initial states. Game boards are randomly initialized, which forces any solver to perform a great deal of exploration from the first step. Another factor that adds complexity to the planning process is the large bonus score that is awarded only when the board is fully cleared. The ability to correctly assess whether the board can be cleared has significant effect on the planner's behavior. Our evaluation shows that *MCTrasnformer* significantly outperforms top-performing methods in a budget-based setting.

## 2 PRELIMINARIES

### 2.1 BACKGROUND – SAMEGAME

**Game rules.** SameGame is a single-player game, played on a rectangular board of height $H$ and width $W$. The board is randomly filled with tiles of $C$ different colors[1]. Two tiles are considered *adjacent* if they are connected either vertically or horizontally. A *block* of tiles is a group of two or more adjacent tiles of the same color. A tile with no adjacent tiles of the same colors is a *singleton*. At each turn, the player selects a single block (one cannot select singletons), which is then removed from the board. When a block is removed, the board is reorganized as follows: a) tiles above the removed tiles "fall down" (a physics-based model); b) when an entire column is removed, all the columns to its right shift left. The game continues until no more blocks exist on the board, i.e., an empty board or one or more singletons.

**Reward function.** The reward is calculated each turn as follows: $(n_i - 2)^2$, where $n_i$ is the size of block chosen by the user at step $i$. If the board is empty when the game concludes, the player receives an additional 1000 points bonus. If tiles remain on the board, a penalty of $\sum_i (n_i - 2)^2$, where $n_i$ is the number of tiles left of color $i$, is exacted. The score is the sum of all the rewards. This setup creates two (potentially conflicting) goals for the player: the first goal is to create blocks that are as large as possible; the second goal is to empty the board, which may require a larger number of steps, as a result of the need to select smaller blocks.

**Complexity.** A SameGame board is defined as *solvable* if a sequence of actions exists so that the board can be emptied. As shown in Schadd et al. (2008)Takes & Kosters (2009), determining whether a randomly-initialized board is solvable is NP-complete. Therefore, finding a sequence of actions that maximizes the score, regardless of whether the board is solvable, is also NP-complete. The main reasons for this difficulty are the game's high branching factor—around 17, initially—and the fact that the length of an average game in our setup is approximately 27.

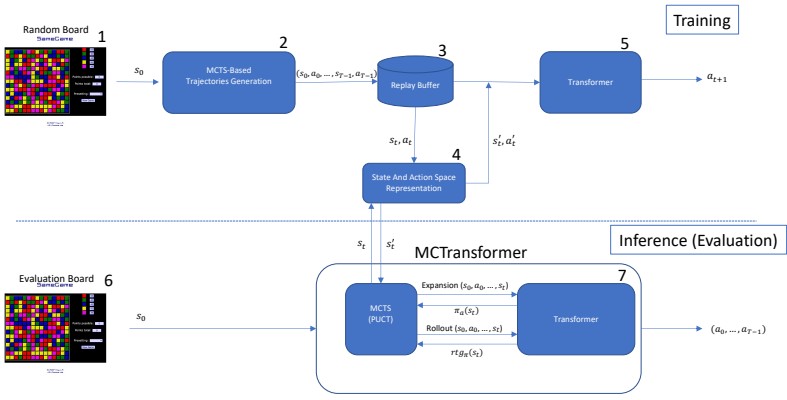

Figure 1: The proposed *MCTrasnformer* approach

## 3 PROPOSED METHOD

### 3.1 OVERVIEW

Our proposed approach is presented in Figure 1. Our training phase (steps #1-#5) is as follows: we begin by generating a batch of randomly-initialized SameGame boards (step #1). Next, for each

---

[1]Our description refers to the version of SameGame used in this study, which is one of the most common.

board we run a fixed number of MCTS simulations (step #2). Each simulation is played until the conclusion of the game, at which point all of its relevant information is saved to the replay buffer (step #3). Next, we use all the games stored in our buffer to generate representations (embeddings) of our states and actions (step #4), which are then used in the training of our Transformer (step #5).

During the test phase, *MCTrasnformer* performs as follows: at each step, our approach uses the MCTS model to select actions until a leaf is reached (i.e., the selection phase). Then, instead of using existing expansion methods, we use our transformer to assign initial scores to each potential action, and then select one based on an upper confidence bound. Next, we use our transformer as a rollout policy: the transformer plays a small set of games to their conclusion, and the scores are used to update values through the MCTS's moves tree.

In addition to its novel architecture, *MCTrasnformer* has important advantages over existing solutions. First, by using transformers to explore and expand the moves tree, we are able to outperform other budget-based methods such as MCTS, as well as transformer-based solutions (as shown in Section 4). Secondly, *MCTrasnformer* requires a smaller number of simulations compared to other methods, a fact that makes it more efficient. Thirdly, our incorporation of the transformer within the MCTS algorithm makes the overall approach more flexible and adaptive: because the transformer learns across multiple board positions, the MCTS does not start as a blank slate with each new board—a highly impactful fact, given that the boards are randomly initialized and have high variance.

### 3.2   Generating MCTS-based Trajectories

The goal of this step is to generate a large volume of games that will enable us to train our Transformer model. We use an MCTS model to generate a large set of games (250,000 in our evaluation), and store the final trajectory of each game in our buffer. Each game begins with a randomly-initialized board, which ensures a diverse set of games. It is important to note that our MCTS model selects the node whose children produced the highest score of each set of simulations, since SameGame is a single-player game that does not involve an adversary. Additionally, we improve the efficiency of the MCTS by unifying all the tiles of a block into a single action, thus significantly reducing the search space. We use the UCT algorithm Matsuzaki (2018) to navigate our tree:

$$UCT(s_t, a) = \overline{Q}_{norm}(s, a) + c_{uct}\sqrt{\frac{\ln N(s_t)}{N(s_t, a)}} \tag{1}$$

where $c_{uct}$ is a hyper-parameter, $N(s_t)$ denotes the number of time the MCTS simulations visit state $s_t$, $N(s_t, a)$ denotes the number of time the MCTS simulations visited state $s_t$ and chose action $a$. Note that $\overline{Q}_{norm}(s, a)$ is defined in Equation 4.

This process is the most time-consuming part of our methods, because of the need to play a large number of games to completion. This difficulty is mitigated by two factors: first, this process takes place during the *offline training* of our model, meaning that it does not affect evaluation time. Secondly, this process can be easily parallelized, since each game can be played independently.

### 3.3   State and Action Space Representations

**The Action Space**. Our action space is represented by a vector whose size is equal to the number of tiles on the board. The values, generated by the Transformer's output layer, are produced using the logits activation function and a softmax operation. It is important to note that while SameGame rules require that we select blocks, our action representation assigns a value to each tile. This is the case for two reasons: first, the number, shapes and sizes of blocks changes throughout the game, and our architecture requires a fixed-size representation. Secondly, a tile-based representation is more nuanced, and provides additional information to the transformer in future time steps. We elaborate on the action selection process, i.e., the transformation of tile-based values to block-based values, in Section 3.4.

**The state space.** Our SameGame board consists of 100 tiles, ordered in a $10x10$ matrix. Our goal for our representation is to capture not only the *color* of each tile, but also the *size* of the block to which it belongs. Therefore, we represent each tile using the following one-hot representation, with

each entry in the vector representing a single color. Note that empty tiles are assigned their own color. The vector representation of each individual tile is in the following form: $(0, 0, ... \frac{s}{n}, 0, ..0)$, where $s$ denotes the size of the tile's block, and $n$ denotes the total number of tiles on the board. This representation maintains the simplicity of a one-hot representation, while also making it easier for our model to identifier larger (i.e., impactful) blocks. Using the one-hot transformation described above, we represent the board as a $H \times W \times (C + 1)$ matrix, where $C$ is the size of the vector representing each tile (i.e., the number of tile colors + background). We then use a convolutional architecture to transform this representation into a vector of size 128.

### 3.4 PRELIMINARY TRAINING THE TRANSFORMER ARCHITECTURE

Our Transformer receives three inputs at each time step: a) the state representation; b) the action representation, and; c) temporal (i.e., positional) encoding that represents the turns of the game. We first use an embedding layer to convert the all three representations to a size of 128. Then, we add (using summation) the temporal encoding *to both* the state and action representations.

The training of our Transformer is performed similarly to that of Transformer-based text generation tasks Chen et al. (2021); Vaswani et al. (2017). Upon receiving a sequence of states and actions $\{s_0, a_o, ..., s_{t-1}, a_{t-1}, s_t\}$, the goal of our model is to predict $a_t$. It is important to note that we only predict actions and not states, because SameGame is a full information, deterministic game, and states can be therefore easily inferred from chosen actions.

One important difference between the MCTS and Transformer components of *MCTrasnformer* lies in the fact that while the former selects blocks, the latter selects tiles. To bridge this gap, we consider the *top-left tile of each block to be the block representative*. This means that when a given block is chosen, the Transformer will be considered to be correct only if the top-left tile of that block is selected. The loss function used by our Transformer is cross entropy.

### 3.5 DEPLOYING *MCTrasnformer* AT TEST TIME

One of the main challenges in efficiently exploring SameGame is the *random initialization of the board*. This setup makes the use of standard MTCS difficult, since the process needs to start 'from scratch' for every new board, which would make the process computationally expensive. Using a fully Transformer-based solution would also be effective, but its inability to effectively balance exploration and exploitation would result in a very long iterative training period (as described in Anthony et al. (2017)).

*MCTrasnformer* combines the strengths of both MCTS and Transformers to create a more efficient solution. The MCTS component manages the exploration/exploitation trade-offs, while the Transformer component is tasked with selecting highly effective actions and optimizing the chosen game trajectories. We begin by using the a PUCT-based version of the MCTS algorithm to reach a leaf in our current search tree. Next, we use the Transformer component to perform the rollout phases. We now describe each phase in detail:

**The selection phase** The goal of this phase is to use a PUCT-based version of MCTS algorithm to reach one of the leaves of the current tree. The selection is carried out the using standard Predictor-UCT Silver et al. (2017):

$$a_t = \arg\max_a (\overline{Q}_{norm}(s_t, a) + U(s_t, a)) \tag{2}$$

where

$$U(s_t, a) = c_{puct} \pi_\theta(s_t, a) \frac{\sqrt{N(s_t)}}{1 + N(s_t, a))} \tag{3}$$

$c_{puct}$ is a hyper-parameter, $\pi_\theta(s_t, a)$ denotes the action probabilities of the stochastic policy $\pi$ parameterized by $\theta$. $N(s_t)$ denotes the number of times state $s_t$ was visited by the MCTS, with $s_t$ being the parent node of node $(s_t, a)$. Similarly, $N(s_t, a)$ is the number of time the MCTS simulations visit state $s_t$ and choose action $a$.

In order to use $UCT$ or $PUCT$ as a selection strategy, the return to go need to be in the range of $[-1, 1]$. In our case, however, the range of return to go has hugh variance and unkown. Therefore,

a normalization method is required. We choose to use Shefi's Seify & Buro (2020) $maxmin$ value normalization method which requires no domain knowledge and no assumption or statistics on the values. This scaling is applied locally at the node level as follows:

$$\overline{Q}_{norm}(s,a) = \frac{2(Q(s,a) - min_{a'}Q(s,a'))}{max_{a'}Q(s,a') - min_{a'}Q(s,a')} - 1 \tag{4}$$

When no action has been taken it is equal to 1, being optimistic. $Q(s,a)$ - the mean of all sum of rewards from state $s$ that took action $a$. $max_{a'}Q(s,a')$ the action $a'$ that return the highest sum of rewards on the trajectory started at state $s$. the same applies for $min_{a'}Q(s,a')$

**The expansion phase.** In this phase, we need to select the next node to be expanded. The selection will be governed by the P-UCT algorithm, with the Transformer setting the initial probabilities of each expansion candidate. For each legal action we can take in the leaf, $\forall a \in actionSpace(s_t)$, we calculate the prior probabilities $\pi_\theta(s_t, a)$. These probabilities are calculated for each action (i.e., block) by *summing the probabilities assigned to all of its tiles.* $\pi_\theta(s_t, a)$ is then used in Equation 3 to select the node that will be expanded.

**The rollout phase.** Once a node has been selected for expansion, we need play the game to its end to obtain a score. Instead of using multiple MCTS simulations, we use *MCTrasnformer*'s Transformer component to play *a single game* to its end. The final score obtained by this run is used to update all the nodes in the trajectory from the MCTS tree's root to the newly expanded leaf.

Following the single rollout run described above, we conduct a small number of additional simulations. These runs start from the root of the tree (not the expanded node), and follow the MCTS algorithm until they reach a leaf. Then, without any additional expansion of nodes, the Transformer component plays the games to their end, and updates the relevant nodes according to the obtained score. We found that this small number of runs (4-16 in our experiments) significantly contribute to the performance of our proposed approach, at a relatively small computational cost. In all rollout runs, the Transformer's policy is deterministic, with the probability of each block obtained by summing the individual probabilities allocated to its tiles.

### 3.6 FINAL ACTION SELECTION

Once all rollouts are completed for the given root, *MCTrasnformer* is faced with the final step of selecting the next move. This selection will be made based on the Q-values of all legal actions (i.e., nodes). It is important to note that this decision only applies to a single step forward, upon the completion of which the MCTS-based process will resume. The root of the MCTS will also be reallocated to the current state of the board.

Our logic in selecting the path with the highest possible Q-value is simple: given that we play a deterministic one-player game, even a single simulation where we are able to obtain a given score ensures that this trajectory can be repeated, and that the obtained value is a lower bound on possible performance.

## 4 EVALUATION

Our evaluation has several goals. Primarily, our aim is to evaluate the efficacy and computational efficiency of *MCTrasnformer* compared to standard Transformer and MCTS-based solutions. Additionally, we will explore the ability of the Transformer-based player to learn the joint distributions of states and actions from our generated datasets, and use several metrics to analyze the perofmance of our proposed appraoch.

### 4.1 EXPERIMENTAL SETUP

**Model architecture.** We use a Transformer architecture with three layers and a single self-attention head. The dimension of each token embedding is 128. We apply Dropout at the end of each decoder block with a probability 0.1. We follow the learning rate scheduling proposed by Radford et al. (2018), with the learning rate increasing linearly from 0 to $6 \times 10^{-4}$ over the course of 3600 updates,

and then using cosine decay for up to 7200 steps. We use a batch size of 64. The full details of the architecture are presented in the Appendix.

**Initial training set.** As described in Section 3.2, we use a MCTS-based model to generate 250,000 games. We use these games to train *MCTrasnformer*'s Transformer component. To generate this dataset, we ran 30 distributed MCTS on random boards, configured with simulation decay and an initial value of 16 simulations per move. Note that a full list of our model's hyperparameters is presented in the Appendix.

**Evaluation datasets.** We evaluate *MCTrasnformer* and the baselines on three data-sets. All datasets consist of boards whose dimensions are $10 \times 10$, and have five colors each. The configurations of the datsets are as follows:

- **100 fully-random boards** – all tile colors are assigned randomly.
- **25 balanced boards** – each board contains exactly 20 tiles of each color. This dataset represent a more difficult planning problem, since the likelihood of creating large blocks is smaller than in cases where one of the colors is dominant.
- **25 dominant-color boards** – one of the five colors consists of more than 40 tiles. This setup consists of a less challenging planning problem, because the chances of creating larger blocks (that will assist in clearing the board) is larger.

**Hardware and training time:** *MCTrasnformer*'s training took place on n1-highmem-8 instance on Google Cloud with 2 NVIDIA Tesla T4 GPUs, for 8 epochs, taking approximately 6 hours.

## 4.2 PERFORMANCE INDICATORS

We record and analyze multiple indicators of the performance of our analyzed methods. The goal is not to only evaluate models based on their final score (although it is the a primary metric), but also based on efficiency and relative efficacy per step. Our metrics are:

1. **Score** $RTG(s_0) = \sum_{t=0}^{T-1} r(s_t, a_t)$. The primary indicator, used to evaluate overall performance.
2. **Net Score.** The sum of all the rewards throughout a game, but **without** the bonus of clearing the board, and also without the penalty for remaining tiles. Specifically, a policy that aims to create large blocks will score higher on this metric.
3. **Final Reward.** The last reward is either a bonus of 1000 points for clearing the board, or a penalty for remaining tiles (see Section section 2.1).
4. **Total Removed Tiles.** The total number of tiles removed in a game. This indicator is equivalent to the score, but the defined reward function (see Section 2.1) can change the policy puts accordingly.
5. **Number Cleared Boards.** - This metric measures the number of games in which the board was cleared. This metric is less noisy that evaluating the overall score and the associated 1000 point bonus.
6. **Average Reward per Move** - $\frac{RTG(S_0)}{T-1}$. An strong model would seek to optimize its moves (i.e., clear blocks that are as large as possible) while also aiming to obtain the bonus. Models that can effectively pursue these two goals will score higher on this metric.
7. **Number of Simulations.** Because our framework is based on MCTS, we choose the number of total simulations per episode as an indicator for the method efficiency. This metric is not relevant for the transformer-based player (see next section).

We report mean and standard deviation for each indicator when applicable.

## 4.3 TRANSFORMER-BASED PLAYER

We begin by evaluating the Transformer-based component of our model. Our goal is to determine whether our Transformer is indeed capable of learning across multiple boards and positions, and select effective moves. The training process of the model is identical to the one described in Section

| | 100 random boards | | 25 balanced boards | | 25 dominant color boards | |
|---|---|---|---|---|---|---|
| | random | transformer | random | transformer | random | transformer |
| Total Score | 3229 | **13517** | 647 | **2600** | 18727 | **29118** |
| Score | 42.2(135) | **135(88.8)** | 25.9(83.6) | **104(61.5)** | 766(356) | **1173(437)** |
| Net. Score | 78.8(58.3) | **155(78.4)** | 25.9(83.6) | **119(52.0)** | 766(356) | **1173(437)** |
| Final Reward | -36.5(117) | **-20.5(24.6)** | -50.7(57.7) | **-14.7(19.6)** | -16.8(21.5) | **-8(12.6)** |
| Removed Tiles | 79.9(9.13) | **84.6(6.74)** | 77.7(7.97) | **86.7(6.68)** | 85.7(6.19) | **88.8(4.92)** |
| Moves | 27.0(3.96) | 26.1(3.01) | 26.0(3.70) | 27.4(3.15) | 18.7(2.99) | **16.6(2.12)** |
| Avg. Reward | 3.08(2.64) | **6.25(3.86)** | 3.06(1.96) | **4.47(2.26)** | 42.9(22) | **72.7(31.3)** |

Table 1: The performance of Transformer-based player against a random player on three sets of boards. All performance indicators except Total Score are provided as mean(std).

3.4. The input of the Transformer is the sequence of all states and actions from the start of the game (i.e., non-Markovian). We evaluate our Transformer in a straightforward manner: at each state, our model receives its input, and then selects an action (the model is deterministic). There are no multiple simulations at any stage, and this step is repeated until the game ends.

Given that our Transformer does not have an exploration/exploitation mechanism, it would require significant amounts of time and computing resources to achieve top performance. This type of evaluation is beyond the scope of this study, which focuses on efficiency. We therefore compare our performance to that of a *random agent*, with the simple goal of demonstrating that our Transformer is capable of meaningful and effective learning. The results of our evaluation are presented in Table 1, and they show that our Transformer significantly outperforms the random baselines in all metrics. The average reward per move metric, for example, shows that the Transformer is much more capable in selecting larger blocks, which also help to drive up its total score.

Finally, we evaluate the Transformer's performance when it only receives the current state as input (i.e., a Markovian model). Our goal is to determine whether not providing the Transformer with the full trajectory (up to the current state) would reduce its performance. Our results, presented in Table 6 in the Appendix, clearly show that the lack of trajectory history significantly reduces performance. For example, the total score for the 100 random boards setup, decreases by 62%, from 13,517 to 5,085. We therefore conclude that providing full trajectory history to our model is required for its performance.

### 4.4 *MCTrasnformer*

We now compare *MCTrasnformer* to various versions of the MCTS algorithm. Because of its ability to balance exploration and exploitation, MCTS is a high-preforming baseline. The quality of its performance, however, is highly-dependent on the number of simulations per move.

Our evaluation is conducted as follows: for the 100 random boards dataset, we evaluate the *MCTrasnformer* with 4, 8, 16 and 20 simulation per move, while the MCTS is evaluated with 8, 16, 32, 64 and 128. Our results, presented in Table 2 (where we only show the x16 configutaion due to space constraints) and Figure 2, shows that *MCTrasnformer* outperforms even the MCTS configuration with the 128 simulations per move. The figure also clearly shows that *MCTrasnformer* forms a Pareto front compared to MCTS (i.e., a better score/number of simulations trade-off at all points). Even more importantly, *MCTrasnformer* leads in three key metrics: total score, net score and average reward/move. These results show that not only does our approach consists of very few dominant boards. Our model therefore has greater difficulty in planning for such scenarios.

## 5 RELATED WORK

### 5.1 THE MONTE-CARLO TREE SEARCH ALGORITHM

Monte Carlo Tree Search (MCTS) is a heuristic search algorithm, that has proven itself highly effective in multiple domains Browne et al. (2012); Swiechowski et al. (2022). The method navigates through a search tree in a manner that balances exploration (evaluating new options) and exploitation (taking advantage of existing knowledge). MCTS is often used as a general-purpose planning

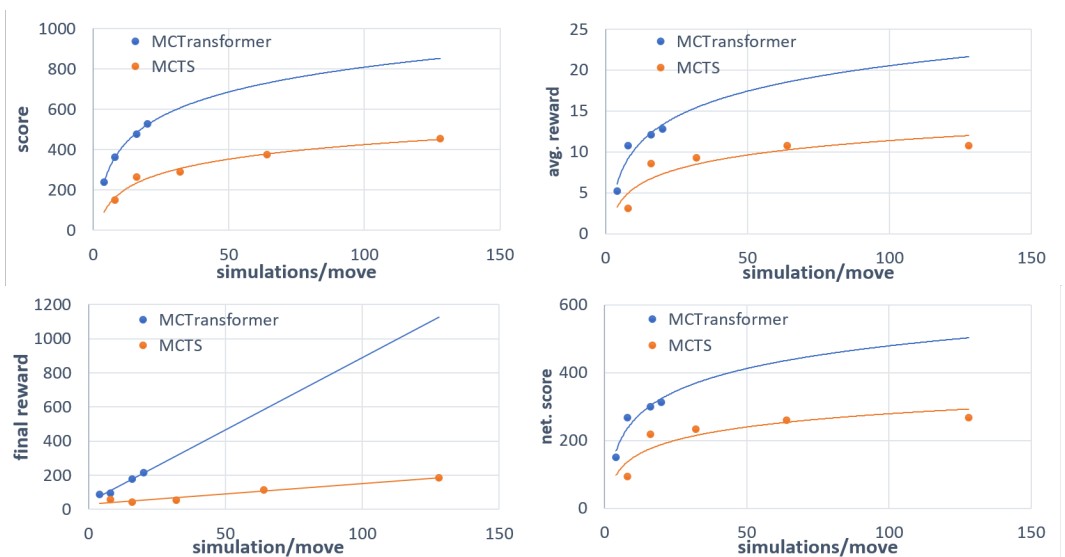

Figure 2: The performance of *MCTrasnformer* against MCTS on 100 random boards as a function of the number of simulations per move.

| model | *MCTrasnformer* x16 | MCTS 8 | MCTS 16 | MCTS 32 | MCTS 64 | MCTS 128 |
|---|---|---|---|---|---|---|
| Total Score | **47704** | 15088 | 26340 | 28951 | 37506 | 45567 |
| Score | **477.04±383.06** | 150.88±250.14 | 263.4±236.19 | 289.51±261.1 | 375.06±388.56 | 455.67±388.56 |
| Net Score | **300.39±127.82** | 94±42.87 | 218.6±114.8 | 229.5±107.84 | 260.29±121.88 | 268.68±109.14 |
| Final Reward | **176.65±387.75** | 56.88±239.62 | 44.81±220.64 | 55.89±239.8 | 114.77±328.6 | 186.99±395.77 |
| Removed Tiles | **92.27±5.5** | 93.52±5.25 | 90.51±5.24 | 90.74±4.78 | 90.29±5.53 | 92.42±5.26 |
| Simulations | 433.28±55.36 | 261.28±25.3 | 446.7±52.5 | 873.6±103.7 | 1711±238 | 3480±410 |
| Moves | 27.08±3.46 | 31.66±3.16 | 26.9±3.29 | 26.3±3.24 | 25.74±3.73 | 26.19±3.2 |
| Cleared Boards | 18 | 6 | 5 | 6 | 12 | **19** |
| Avg. Reward | **12.13±6.36** | 3.085±1.68 | 8.56±5.48 | 9.3±5.03 | 10.73±6.13 | 10.78±5.68 |

Table 2: *MCTrasnformer* with 16 simulations per move compared to MCTS on on 100 random boards

algorithm, and it was effectively applied to diverse domains such as Bayesian reinforcement learning Vien et al. (2013), multi-party negotiations Golpayegani et al. (2016), combinatorial optimization Ontanón (2013), and sigle-player games Schadd et al. (2008); Seify & Buro (2020); Rosin (2011); Méhat & Cazenave (2010). More recently, MCTS has been an important part of the highly influential AlphaGo Silver et al. (2016) and AlphaZero Silver et al. (2017) architectures.

MTCS-based solutions have several important strengths. First, the approach does not require domain-specific knowledge Klein (2015) Jacobsen et al. (2014) Hu et al. (2019), and thus it is easily applied to any domain that can be represented as set of sequential decisions. Secondly, while heuristics and prior knowledge are not required for it to perform well, MCTS can be augmented with such techniques to improve its ability to assess alternative actions. Thirdly, the training of the model can be performed on any available budget; once the budget is exhausted, the MCTS can utilize whatever knowledge it has been able to obtain.

## 5.2 UTILIZING TRANSFORMERS IN REINFORCEMENT LEARNING

Despite their state-of-the-art performance in many domains, Transformers have only been recently introduced to the field of RL. The first attempts in this field sought to adapt Transformers so that they can be used within existing solutions. In Parisotto et al. (2020), the authors present the Gated Transformer architecture. By integrating a gating mechanism into the original transformer architecture, the authors were able to improve the model's stability and successfully utilize it for RL tasks.

| model | *MCTrasnformer* x8 | MCTS 8 | MCTS 16 | MCTS 32 | MCTS 64 | MCTS 128 |
|---|---|---|---|---|---|---|
| Total Score | **12173** | 3944 | 5516 | 6084 | 6908 | 11193 |
| Score | **486.92±422.53** | 157.76±76.99 | 220.64±232.18 | 243.36±210.88 | 276.32±215.71 | 447.72±394.58 |
| Net Score | 249.08±86.48 | 161.56±77.6 | 184±73.2 | 207.16±82.7 | 241.52±78.6 | 249.8±76.64 |
| Final Reward | **237.84±437.14** | -3.8±6.27 | 36.6±200.82 | 36.2±200.92 | 34.8±201.25 | 197.92±409.33 |
| Removed Tiles | 93.32±5.38 | 90.68±4.14 | 91.12±4.15 | 90.6±2.92 | 90.64±4.46 | 93.72±4.77 |
| Simulations | 219.52±27.24 | 226.56±28.72 | 449.92±40.08 | 871.68±93.42 | 1725.44±215.84 | 3481.6±419.68 |
| Moves | 26.44±3.4 | 27.32±3.59 | 27.12±2.5 | 26.24±2.92 | 25.96±3.37 | 26.2±3.28 |
| Cleared Boards | **6** | 0 | 1 | 1 | 1 | 5 |
| Avg. Reward | 9.83±4.25 | 6.29±3.66 | 7.02±3.3 | 8.13±3.62 | 9.67±3.9 | 9.88±3.82 |

Table 3: *MCTrasnformer* with 8 simulations per move compared to MCTS on 25 balanced boards

| model | *MCTrasnformer* x8 | MCTS 8 | MCTS 16 | MCTS 32 | MCTS 64 | MCTS 128 |
|---|---|---|---|---|---|---|
| Total Score | 42062 | 42880 | 44209 | 46435 | 50156 | **52584** |
| Score | 1682.48±426.74 | 1715.2±502.82 | 1768.36±491.37 | 1857.4±525.2 | **2006.24±553.15** | **2103.36±483.51** |
| Net Score | 1602.28±474.96 | 1640.96±522.9 | 1731.64±499.52 | **1823.4±517.87** | 1809.88±499.66 | 1745.2±571.99 |
| Final Reward | 76.2±278.09 | 74.24±279 | 34±201.49 | **110.79** | 196.36±410.18 | 358.16±491.32 |
| Removed Tiles | 91.24±5.07 | 91.36±5.75 | 91.96±5.07 | 90.68±5.65 | 92.76±5.84 | **95.2±5.05** |
| Simulations | 136.64±24.32 | 226.56±28.72 | 290.56±40.96 | 556.8±75.05 | 1192.96±168.19 | 2447.36±508.41 |
| Depth | 16.08±3.04 | 17.36±2.94 | 17.16±2.56 | 16.4±2.35 | 17.64±2.63 | 18.12±3.97 |
| Cleared Boards | 2 | 2 | 1 | 1 | 5 | **6** |
| Reward | 105.07±39.2 | 99.98±41.43 | 107.4±40.74 | 116.67±45.17 | 107.4±40.7 | 104.88±48.04 |

Table 4: *MCTrasnformer* with 8 simulations per move compared to MCTS on 25 dominant color boards

In Zambaldi et al. (2018), Transformers and other advanced architectures were using to better model advanced relationships in the problem domain.

Instead of integrating them in existing RL frameworks, other studies seek to develop new ways to utilize the strengths of Transformers. In Chen et al. (2021), the authors present the Decision Transformer, whose goal is to model the planning problem as one of autoregressive sequence completion. The model receives as input the state and action representations, as well as the sum of remaining future rewards. The Transformer is then tasked with completing the sequence of states and actions in a way that obtains the specified reward. Another sequence modeling-based approach was proposed in Janner et al. (2021), where instead of future rewards, the model receives the rewards incurred by its current actions. Another difference between the two approaches is the use of Beam Search Zhuo et al. (2020) to improve the planning process.

One of the main challenges in the modeling of RL problems as those of sequence completion/generation, is the difficulty of performing efficient exploration. While exploration is an integral part of many RL algorithms (e.g., MCTS) This is not the case here. As a result, using Transformers to discover effective solutions may require very long training times and be computationally prohibitive. In Zheng et al. (2022b), the authors seek to mitigate this problem by proposing an online training phase, where the Transformer can be fine-tunes to respond to changing circumstances. Another approach was presented in Correia & Alexandre (2022), where a hierarchical Transformer with sub-goals was used to improve the exploration process.

## 6   CONCLUSIONS

One of the main challenges in using Transformers in the field RL is the need to perform effective exploration and planning. Studies that seek to utilize the strengths of the architecture use model the RL as one of sequence modeling, i.e., supervised learning. This setup makes exploration, a key component of RL, more difficult. In this study we proposed a novel approach for integrating Transformers into the MCTS algorithm. This combination enables us to use the latter for effective exploration, while utilizing the former's advanced modeling abilities to identify high-reward paths of action. Evaluation on the challenging domain of SameGame demonstrates that our approach outperforms both the MCTS and Transformer algorithms.

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

# A  APPENDIX

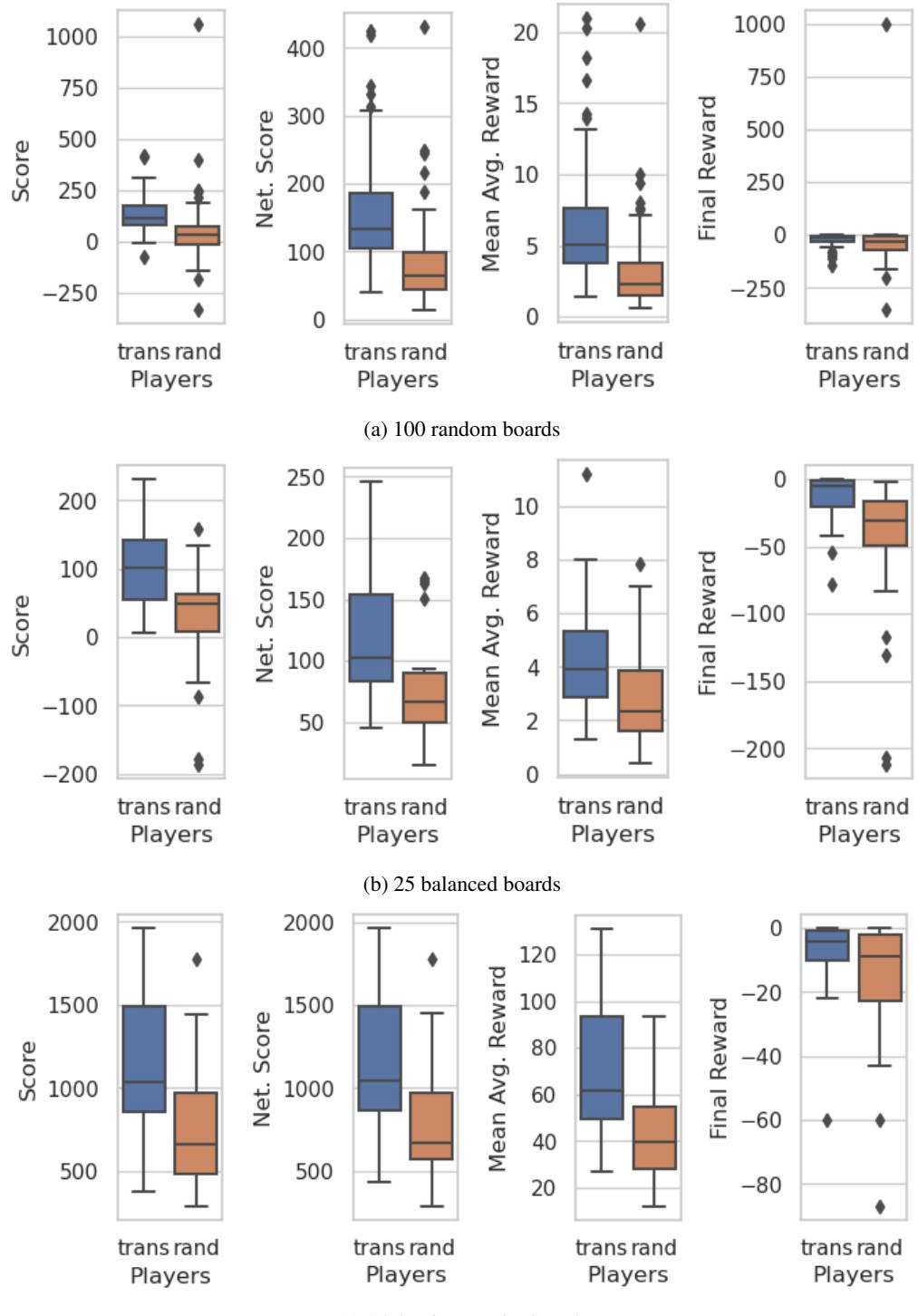

(a) 100 random boards

(b) 25 balanced boards

(c) 25 dominant color boards

Figure 3: The performance of Transformer-based player against a random player on three sets of boards

Table 5: Hyperparameters of Transformer

| Hyperparameter | Value |
| --- | --- |
| Number of layers | 3 |
| Number of attention heads | 1 |
| Embedding dimension | 128 |
| Batch size | 64 |
| Max sequence length | 45 |
| Activation Functions | ELU, encoder, state embedding |
| | GeLU, otherwise |
| | tanh, embedding |
| State embedding channels | 64 |
| State embedding filter sizes | $(3, 3)$ |
| State embedding strides | 1 |
| Epochs | 8 |
| Dropout | 0.1 |
| Learning Rate | $6 \times 10^{-4}$ |
| Adam betas | $(0.9, 0.95)$ |
| Weight decay | 0.1 |
| Learning rate decay | Linear warmup(3600) and cosine decay |

| KPI | transformer |
| --- | --- |
| Total Score | 5085 |
| Score | 50.85±63.1 |
| Net. Score | 82.73±45.75 |
| Final Reward | -31.88±33.99 |
| Removed Tiles | 82.09±7.64 |
| Moves | 25.89±3.35 |
| Avg. Reward | 3.38±2.36 |

Table 6: Transformer performance on 100 random boards with **no attention**

