# OpenReview forum: "MCTransformer: Combining Transformers And Monte-Carlo Tree Search For Offline Reinforcement Learning"
_ICLR.cc/2023/Conference — Submitted to ICLR 2023_

### Official Review · Reviewer_scYv · 2022-10-23

**Confidence:** 4
**Correctness:** 3
**Technical Novelty And Significance:** 2
**Empirical Novelty And Significance:** 2
**Recommendation:** 3

**Clarity, Quality, Novelty And Reproducibility:**

**Clarity**
Although there are many typos and grammar errors in the paper, the presentation is easy to follow.

**Quality & Novelty**
The work is technically correct, however, this work is simply a combination of Transformers and MCTs with limited novelty. Some additional experiments to justify the use of Transformer architecture in MCTS and offline reinforcement learning are needed.

**Reproducibility**
The paper fully provides the information to reproduce the method and to obtain the results as reported, possibly with some effort.


**Strength And Weaknesses:**

**Strength:**
The main idea of this paper is to combine Monte-Carlo Tree Search (MCTS) with the Transformer architecture in offline reinforcement learning.

**Weakness:**
- The method seems simply a combination of existing solutions (MCTS and Transformer) with limited novelty. Readers do not learn much from it.
- Why does this paper choose the game of SameGame only to demonstrate their method? How about 2-player games?
- In addition, the experiments are not convincing for the following. The authors only compare their method (a combination of Transformer and MCTS) with the random player, the Transformer-only and the MCTS-only. As a combination of existing solutions, the authors need to compare it with many other methods to support their claims. For example, those in Schadd et al. (2008); Takes & Kosters (2009); Schadd et al. (2012); Cazenave (2016); Seify (2020); Cazenave(2020). Even, it is probably more critical to compare with AlphaZero. More are listed as follows.
  - To evaluate the effectiveness of MCTransformer on the SameGame, the authors should consider evaluating the proposed method with a standardized test set of 20 positions (these positions can be found at www.js-games.de/eng/games/samegame), which were reported in many prior works such as Schadd et al. (2008); Takes & Kosters (2009); Schadd et al. (2012); Cazenave (2016); Seify (2020); Cazenave(2020).
  - Since the authors use offline data to train the Transformer policy, the author should compare the performance to other offline RL methods, or imitation learning methods (e.g behavior cloning).
  - To show the effectiveness of using Transformer architecture in controlling the exploration and evaluation, the authors should compare the performance of using a Transformer architecture vs other deep neural network architecture.
  - To show the effectiveness of the Transformer to carry out the simulation phase (i.e., the rollout policy) in MCTS, the authors should consider comparing the Transformer to heuristic-based policies or other learning-based policies.
  - Since the author claims the effectiveness of using Transformer as the rollout policy, there is a need to describe the policy in the rollout phase of the MCTS-based method. For example: Is the rollout policy of the MCTS-based is a random policy or heuristic? And how many rollout simulations were performed at the selected leaves?

**Comments for presentation:**
- “Our approach combines Monte-Carlo Tree Search (MCTS) with the Transformer architecture in an actor-critic setup” is not well-explained in the paper.
- Need more discussion for Table 3 and  Table 4.
- Increase the font size of labels in Figure 1.
- There are some typo mistakes and missing punctuations in the paper. For example: in the first line on page 5, the authors mentioned “Shefi’s Seify & Buro (2022) maxmin value normalization method”. Should “Shefi’s” be “Seify’s”?


**References**

[1]The positions can be found at www.js-games.de/eng/games/samegame.
 Maarten PD Schadd, et al. Single-player Monte-Carlo tree search for SameGame. Knowledge-Based Systems 34, 2012

[2] Cazenave Tristan. Nested rollout policy adaptation with selective policies. Computer Games. Springer, Cham, 2016.

[3] Cazenave Tristan, et al. Stabilized nested rollout policy adaptation. Monte Carlo Search International Workshop. Springer, Cham, 2020.


**Summary Of The Paper:**

The paper proposed MCTransformer, a framework that combines Monte-Carlo Tree Search (MCTS) with the Transformer architecture in offline reinforcement learning.
The MCTS component is responsible for navigating and balancing the exploration/exploitation trade-off, while the Transformer component is responsible for evaluating the new states and also serves as a rollout policy to carry out the simulation phase of the MCTS.
The evaluation on SameGame shows that MCTrasnformer outscores Transformer-only and MCTS-only solutions.


**Summary Of The Review:**

As described in the weaknesses above, it is hard to recommend this paper.

---

### Official Review · Reviewer_cK37 · 2022-10-25

**Confidence:** 5
**Correctness:** 3
**Technical Novelty And Significance:** 1
**Empirical Novelty And Significance:** 1
**Recommendation:** 1

**Clarity, Quality, Novelty And Reproducibility:**

The paper is easy to follow. The quality and novelty are poor. The author did not provide the source code for reproducibility. However, since the method is similar to AlphaZero, it is quite easy to reproduce it.

**Strength And Weaknesses:**

The main concern for this paper is the novelty. The authors simply used supervised learning to train the transformer network from the data generated by the MCTS-Based program on SameGame. Then, they apply the transformer network to MCTS. The structure is identical to AlphaGo/AlphaZero’s setting except only replacing the neural network from Resnet to Transformer.

Besides, the experiment is unfair and not convincing:
- Without using neural networks, the MCTS-based program can execute more simulation counts. The author should provide more experiment results to show that the comparison is fair, e.g. comparing the experiment with the same time limit rather than simulation counts.
- It is very common to combine the MCTS with neural networks, such as AlphaGo, AlphaZero, or MuZero. The authors should compare the MCTransformer to other MCTS-based programs with neural networks.

Other minor issues:
- The author should consider using MCTransformer to generate the training dataset (step#2 in Figure1) as AlphaZero did. Currently, the performance of MCTransformer may be bound by the MCTS-based program.
- In Figure 1: The definition of the step#6 and step#7 are unclear.
- In section 3.5, “... standard MTCS difficult … ” should be “... standard MCTS difficult …”.
- In section 5.1, “MTCS-based solutions …” should be “MCTS-based solutions …”.
- Does it require a lot of time for using the Transformer in the rollout?
- It’s meaningless to compare a well-trained network to a random play in Table 1.
- The authors should try more simulation counts in the experiment in Figure 2 instead of using the regression line.


**Summary Of The Paper:**

This paper proposes MCTransformer by incorporating Transformer into Monte-Carlo Tree Search. The MCTransformer is then applied to SameGame, and the experiments show that MCTransformer outperforms both Transformer-only and MCTS-only solutions.


**Summary Of The Review:**

The novelty of this paper is very poor. The structure is similar to AlphaGo/AlphaZero’s setting except only replacing the neural network from Resnet to Transformer. Overall, I did not find any new results in this paper.

---

### Official Review · Reviewer_XF4h · 2022-10-30

**Confidence:** 4
**Correctness:** 2
**Technical Novelty And Significance:** 3
**Empirical Novelty And Significance:** 2
**Recommendation:** 3

**Clarity, Quality, Novelty And Reproducibility:**

Clarity:
  * The paper lacks clarity in exposition, and requires multiple passes to grok and connect ideas.
  * Figure 1 should be self-explanatory with an elaborate caption. Nits: The font-sizes in the figure are extremely small. Some things like the rollout component in the figure are left unexplained (what's $rtg_\pi(s_t)$?)
  * The MCTS baseline needs more explanation as well.
  * Consider using `citep` for inline citations so they land inside brackets.
  * The word transformers is used very liberally, but remember that transformers as just an architecture and can be used in RL in a number of ways. This paper uses transformers to do sequential modelling of trajectories, which should be explicit rather than implicit. Consider replacing transformers by decision transformers or trajectory transformers to avoid the ambiguity.

Quality:
  * The paper starts off with a great idea, but lacks in execution and clarity.

Originality:
 * The idea and method itself are pretty original to my knowledge.

**Strength And Weaknesses:**

Strengths:
* The paper tries to fix a key issue with Decision Transformers (DTs; transformers for sequence modelling of RL trajectories) like methods which is that they can't adapt easily at test time to explore more or improve their performance.
* The particular setup of combining DTs and MCTS is novel, and I could see follow-up working building further on it.

Weaknesses:
* The writing and presentation is the paper is extremely unclear. The exposition of the method could be a lot better. See the below section for some suggestions.
* The evaluation is extremely limited and non-standard, both in terms of the domains and algorithms. This makes is hard to assess if the original pitch of the paper empirically holds.
  * The experiments performed are on a single, non-standard domain of SameGame which is relatively toy-ish as well. I would have liked seeing comparisons on more well-benchmarked domains like MiniGo or Atari. Similarly, there were no comparisons to DecisionTransformers or Online DTs which the paper supposedly improves upon.
  * I am also not sure what to make of the MCTS comparisons, since from my understanding the paper doesn't use a learned prior policy or value function in the MCTS experiments (similar to AG/AGZ/Muzero), which makes the comparisons a bit unfair IMO.



**Summary Of The Paper:**

This work proposes to combine Decision/Trajectory Transformers (DTs) with MCTS, in order to improve the performance of DTs and also enable better exploration during online evaluation. The paper uses transformers as a prior policy for MCTS during the expansion phase, and also uses the same transformer to perform rollouts. The experiments are performed in the SameGame domain, which is a fully-observable Tetris like game with state-based inputs. The experiments show benefit over using using vanilla MCTS that learns only from monte-carlo returns.

**Summary Of The Review:**

This paper proposes an elegant idea to combine MCTS with Trajectory Transformers to enable the latter to perform online improvement / exploration. While the idea itself is promising, the paper lacks in empirical execution and evidence (it terms of the domains and compared algorithms). The writing and exposition itself need some work. Owing to these factors, I don't think the paper is ready for acceptance at this time, but I do see a lot of legroom in the idea itself so this could be a great paper once the issues are fixed.

---

### Decision · Program_Chairs · 2023-01-20

**Decision:**

Reject

**Justification For Why Not Higher Score:**

Limited novelty and experiment support.

**Justification For Why Not Lower Score:**

N/A

**Metareview: Summary, Strengths And Weaknesses:**

The work proposes MCTransformer, a framework that combines Monte-Carlo Tree Search (MCTS) with Transformers, to introduce exploration into transformer based offline RL approaches. The algorithm is evaluated on SameGame and showed improved performance over baselines. All the reviewers expressed concerns regarding novelty of the work, i.e., combination of existing solutions (MCTS and Transformer) with limited novelty, as well as limited experimental support, e.g., single domain, missing baselines (MCTS combined with other neural networks, other offline RL SoTA).